# Urine and Serum miRNA Signatures for the Non-Invasive Diagnosis of Adenomyosis: A Machine Learning-Based Pilot Study

**DOI:** 10.3390/diagnostics15233012

**Published:** 2025-11-26

**Authors:** Tomas Kupec, Julia Wittenborn, Chao-Chung Kuo, Rebecca Senger, Philipp Meyer-Wilmes, Laila Najjari, Elmar Stickeler, Jochen Maurer

**Affiliations:** 1Clinic for Gynecology and Obstetrics, University Hospital RWTH Aachen, Pauwelsstraße 30, 52074 Aachen, Germany; 2Genomics Facility, Interdisciplinary Center for Clinical Research (IZKF), University Hospital RWTH Aachen, Pauwelsstraße 30, 52074 Aachen, Germany

**Keywords:** adenomyosis, endometriosis, liquid biopsy, serum miRNA, urine miRNA, biomarkers, non-invasive diagnosis, machine learning

## Abstract

**Background:** Adenomyosis remains difficult to diagnose non-invasively due to clinical overlap with endometriosis and the limited specificity of imaging techniques. This pilot study evaluated whether serum- and urine-derived microRNA (miRNA) profiles, combined with machine-learning approaches, could support non-invasive diagnosis. **Methods:** Serum and urine samples were collected from 59 patients undergoing surgery for chronic pelvic pain at the Endometriosis Centre of RWTH Aachen University Hospital. Seven patients had isolated adenomyosis, 34 had histologically confirmed endometriosis, and 18 served as negative controls. miRNAs were profiled using next-generation sequencing. A structured feature-selection pipeline (variance filtering, univariate testing, mutual information, recursive feature elimination) was applied before training Logistic Regression, Random Forest, Support Vector Machine, and Decision Tree models using cross-validation. Model performance was evaluated using accuracy, precision, recall, F1 score, and ROC-AUC. **Results:** Distinct miRNA signatures were detected in both serum and urine, with urine-based models showing superior discriminatory performance. Logistic Regression and Support Vector Machine achieved excellent separation in urine datasets, although perfect AUC values must be interpreted cautiously due to the small number of adenomyosis cases. In serum, Random Forest achieved the highest AUC values (up to 0.98). Several miRNAs, including miR-183-3p, miR-320d-2, and miR-17, emerged as promising candidate biomarkers for differentiating adenomyosis from endometriosis and from negative controls. **Conclusions:** This pilot study demonstrates the feasibility of liquid-biopsy miRNA profiling combined with machine learning for non-invasive adenomyosis detection. Although results are preliminary and require validation in larger cohorts, urine miRNA profiles may represent a promising complementary tool to improve diagnostic accuracy and reduce diagnostic delay.

## 1. Introduction

Adenomyosis is a chronic gynecological disorder characterized by the ectopic presence of endometrial glands and stroma within the myometrium commonly leading to dysmenorrhea, menorrhagia and chronic pelvic pain [1]. Reported prevalence among hysterectomy patients has varied widely from 8.8% to 61.5% over the past five decades, reflecting diagnostic heterogeneity and variation in study population [2]. Although adenomyosis frequently coexists with endometriosis, it represents a distinct pathological entity with specific clinical, biological and therapeutic challenges [3]. Co-occurrence rates range between 36% and 62% [4], and the substantial symptom overlap complicates differential diagnosis, underscoring the need for improved, non-invasive biomarkers capable of supporting accurate patient stratification and reducing unnecessary surgical procedures.

The diagnosis of adenomyosis relies primarily on transvaginal ultrasound (TVUS) and magnetic resonance imaging (MRI). The Morphological Uterus Sonographic Assessment (MUSA) criteria provide standardized sonographic definitions of direct and indirect features of adenomyosis [5,6]. TVUS demonstrates sensitivities of 75–83.8% and specificities of 63.9–81% [7,8] I achieves sensitivities of 69–77% and specificities of 80–89% [8,9,10]. Despite these advances, imaging remains operator-dependent and moderately accurate, and adenomyosis continues to be underdiagnosed, particularly in young patients or those with coexisting endometriosis.

Beyond diagnostic challenges, adenomyosis is associated with impaired reproductive outcomes, including reduced pregnancy rates, higher miscarriage risk, preterm birth, premature rupture of membranes, and fetal growth restriction [11,12,13]. These observations highlight the importance of earlier and more precise diagnosis to optimize clinical management and fertility counseling.

MicroRNAs (miRNAs) are small, non-coding RNA molecules involved in post-transcriptional gene regulation [14]. Dysregulated miRNAs contribute to key pathological mechanisms implicated in adenomyosis—including inflammation, epithelial–mesenchymal transition (EMT), fibrosis, aberrant proliferation, and cytoskeletal remodeling [15,16,17]. Recent studies have mapped adenomyosis-specific miRNA alterations, including recurrent dysregulation of miR-10b, miR-30c-5p, miR-21, miR-145, and others that regulate critical pathways such as PI3K–AKT/mTOR, MAPK, FAK/Talin1, and EMT-associated transcription factors [18]. These processes underpin invasive behavior at the endometrial–myometrial interface and stromal cell survival—hallmarks of adenomyosis pathophysiology. miRNAs are highly stable in circulating biofluids and can be robustly detected in serum, urine, saliva, and peritoneal fluid [19,20,21]. Prior work, including our own studies, has demonstrated excellent stability of circulating miRNAs in serum and urine and their suitability for minimally invasive diagnostic applications [22,23]. While miRNA-based diagnostics have been investigated extensively in endometriosis and gynecologic malignancies, translational research in adenomyosis remains comparatively limited, and no validated miRNA panels are currently available for clinical use.

Machine learning (ML) techniques offer new opportunities for identifying complex, multidimensional biomarker signatures from high-dimensional sequencing data. Recent evidence from large-scale endometriosis studies has shown that ML-derived miRNA signatures can achieve high diagnostic accuracy across independent validation cohorts [21,24,25]. However, comparable ML-based approaches have not yet been systematically applied to adenomyosis.

Building on these emerging concepts, the present pilot study aimed to identify candidate miRNA biomarkers for adenomyosis using next-generation sequencing of serum and urine samples from patients presenting with chronic lower abdominal pain at the specialized Endometriosis Centre of RWTH Aachen University Hospital. By implementing a structured ML pipeline—comprising variance filtering, univariate testing, mutual information, recursive feature elimination, and multi-algorithm classification modeling—we sought to determine whether adenomyosis is associated with distinct biofluid-specific miRNA expression patterns and whether these signatures can discriminate adenomyosis from endometriosis and from negative controls. The findings from this study provide preliminary evidence supporting the feasibility of miRNA-based, non-invasive diagnostic strategies for adenomyosis and lay the groundwork for future validation in larger, independent cohorts.

## 2. Materials and Methods

### 2.1. Patient Selection

This prospective study included 59 patients who were evaluated for lower abdominal pain or suspected endometriosis at the Endometriosis Centre of RWTH Aachen University Hospital between December 2021 and August 2023. Among these, 7 patients were diagnosed with adenomyosis as a single diagnosis, 34 patients had histologically confirmed endometriosis and served as positive controls, and 18 patients in whom endometriosis and adenomyosis were excluded intraoperatively served as negative controls.

Before clinical examination, all patients completed a standardized medical history questionnaire documenting demographic characteristics and self-reported symptoms. The diagnosis of endometriosis was made in accordance with the current ESHRE guidelines [26], based on gynecological examination and transvaginal ultrasound performed by an experienced senior consultant at the specialized Endometriosis Centre. Consultations routinely included individualized treatment planning, including endocrine therapy, surgery, analgesia, reproductive counseling, and supportive care.

Eligibility criteria comprised all patients scheduled for diagnostic or therapeutic laparoscopy during the study period due to clinical suspicion of endometriosis based on chronic lower abdominal pain or related symptoms. All surgical procedures were performed by one of three highly experienced surgeons at the Endometriosis Centre. During laparoscopy, the entire abdomen and pelvis were systematically inspected. Any endometriotic lesions were excised and subjected to histopathological confirmation. All patients in the positive control group had histologically confirmed endometriosis.

The negative control group consisted of patients who underwent surgery for suspected endometriosis but in whom endometriosis and adenomyosis were excluded intraoperatively. Adenomyosis was diagnosed through a combination of preoperative transvaginal ultrasound (TVUS) and intraoperative assessment. For TVUS diagnosis, we applied the MUSA criteria as refined by Harmsen et al. [6]. These criteria include direct signs of adenomyosis—myometrial cysts, hyperechogenic islands, echogenic subendometrial lines and buds—and indirect signs such as asymmetric myometrial wall thickening, fan-shaped shadowing, a globular uterine configuration, translesional vascularity, and irregular or interrupted junctional zone features.

All participants provided written informed consent prior to inclusion. The study was conducted in accordance with the Declaration of Helsinki and approved by the Independent Ethics Committee of the Faculty of Medicine at RWTH Aachen University (ethics vote 206/09).

### 2.2. Sampling

Serum

Before surgery, serum samples were collected as part of the routine preoperative laboratory examination for miRNA analysis. A total of fifty-nine serum tubes (Greiner Bio-One, Kremsmünster, Austria), each containing 10 mL of whole blood, were obtained. After collection, the tubes were stored at 4 °C at the Endometriosis Centre until complete clot formation. Samples were then centrifuged at 2500× *g* for 10 min. The serum supernatant was carefully pipetted, aliquoted, and stored at −80 °C on the same day. All samples were registered in the biobank of RWTH Aachen University Hospital and labeled using pseudonymized identifiers prior to further processing.

Urine

Before surgery, urine samples were collected as part of the routine preoperative laboratory examination for miRNA analysis. A total of fifty-nine urine tubes (Greiner Bio-One, Kremsmünster, Austria), each containing 20 mL of urine, were obtained. After collection, the samples were stored at 4 °C at the Endometriosis Centre. On the same day, all samples were centrifuged at 944× *g* for 10 min at room temperature to separate the cell pellet. The cell-free supernatant was carefully pipetted, aliquoted, and subsequently stored at −80 °C. All samples were registered in the biobank of RWTH Aachen University Hospital and labeled with pseudonymized identifiers prior to further processing.

Sample preparation

For miRNA extraction, at least 7 mL of cell-free supernatant was available per sample; of this, 4 mL was used for extraction, and the remaining volume was retained as backup material. In total, samples from fifty-nine patients (*n* = 34 endometriosis patients as positive controls, *n* = 7 patients with adenomyosis, and *n* = 18 negative controls) were included in the final analysis. miRNA was isolated from each serum and urine sample using the miRNeasy Serum/Urine Kit (Qiagen, Hilden, Germany) according to the manufacturer’s instructions.

Sequencing libraries were prepared using the QIASeq miRNA UDI Library Kit (Qiagen, Hilden, Germany) according to the manufacturer’s instructions. For each biofluid sample, the recommended 4 µL input volume was supplemented with 1 µL of synthetic miRNAs from the QIASeq miRNA Library QC Kit to serve as an internal quality control. Library quality was assessed using a Bioanalyzer or TapeStation system (Agilent, Waldbronn, Germany), and library concentrations were quantified using a Quantus fluorometer (Promega, Madison, WI, USA). All libraries were sequenced on an Illumina NextSeq 500 instrument (Illumina, San Diego, CA, USA) in 72 bp single-end mode, generating a mean sequencing depth of approximately five million reads per sample.

FASTQ files were generated using bcl2fastq (Illumina NextSeq 500). To ensure reproducibility, raw sequencing data were processed using the publicly available nf-core/smRNAseq pipeline (version 2.3.0) [27], implemented in Nextflow version 23.10.1 [28] and executed using Docker 24.0.2. Downstream analyses were performed in R version 4.3.3 using the DESeq2 framework (v1.38.3) [29] for normalization and count transformation.

All samples (34 positive controls, 7 adenomyosis cases, and 18 negative controls) were included in the downstream computational analyses. After DESeq2 normalization, five-fold cross-validation was applied to ensure robust model performance. Two analytical strategies were implemented: (i) classical ML models (logistic regression, decision tree, random forest, and support vector machine) were trained on the full set of >4000 detected miRNAs to capture broad expression patterns, and (ii) model-based feature selection procedures were applied to identify the most informative miRNAs for classification. All workflows were implemented in Python tools (version 3.12) using scikit-learn, NumPy, Pandas, Matplotlib, and Seaborn.

GraphPad Prism 10 was used for statistical evaluation. Student’s *t*-test was applied to determine significant differences in miRNA expression.

### 2.3. Preprocessing

The initial dataset contained 4285 detected miRNA features in each comparison group, comprising 25 samples for serum adenomyosis vs. negative controls, 41 samples for serum adenomyosis vs. positive controls, 25 samples for urine adenomyosis vs. negative controls, and 41 samples for urine adenomyosis vs. positive controls. Preprocessing steps included the removal of all-zero features, reducing the feature counts to 3407 (serum: adenomyosis vs. negative controls), 3742 (serum: adenomyosis vs. positive controls), 3780 (urine: adenomyosis vs. negative controls), and 4006 (urine: adenomyosis vs. positive controls). This removal of non-informative features ensured that only truly expressed miRNAs were retained for subsequent analyses, thereby reducing noise and improving the efficiency and reliability of the feature selection and classification workflows.

### 2.4. Feature Selection

Feature selection was a crucial step to optimize the classification models by reducing dimensionality and retaining only the most informative miRNA features. A structured four-step feature selection pipeline was applied separately to each dataset (serum and urine; adenomyosis vs. negative controls and adenomyosis vs. positive controls).

First, variance thresholding was performed to remove features with low variance across samples, as these were considered non-informative or redundant. After this step, feature numbers were reduced from 3407 to 2126 (serum: adenomyosis vs. negative controls), from 3742 to 2154 (serum: adenomyosis vs. positive controls), from 3780 to 3313 (urine: adenomyosis vs. negative controls), and from 4006 to 3543 (urine: adenomyosis vs. positive controls).

Second, univariate feature selection was applied to retain features with the strongest individual statistical association with the target variable, further reducing the feature counts to 1063 and 1077 in the serum datasets, and to 1656 and 1771 in the urine datasets (negative and positive controls, respectively).

Third, mutual information filtering was used to capture both linear and non-linear dependencies between miRNAs and class labels. This step reduced the serum feature sets to 531 (negative controls) and 538 (positive controls), and the urine feature sets to 828 and 885 features, respectively.

Finally, Recursive Feature Elimination (RFE) was conducted to identify the most relevant predictors. RFE iteratively removed the least informative features based on supervised model training, ultimately selecting the top 20 most predictive miRNAs for each classification task. This comprehensive approach reduced overfitting, improved model stability, and ensured that the final feature sets consisted of biologically meaningful candidate biomarkers.

### 2.5. Model Selection and Evaluation

To ensure robust and unbiased evaluation of model performance, each dataset was divided into training and test sets. For comparisons involving adenomyosis versus negative controls (serum and urine), the data were split into 15 training and 10 test samples. For comparisons against positive controls, 24 samples were allocated to training and 17 to testing in both biofluids. To further enhance generalizability and reduce sampling bias, K-fold cross-validation was applied during the training phase across all models.

Four classification algorithms were developed and tested:Logistic Regression;Decision Tree;Random Forest;Support Vector Machine (SVM).

Model performance was assessed using multiple complementary metrics, including accuracy, precision, recall, F1 score, and the area under the ROC curve (AUC). This comprehensive evaluation framework allowed a detailed comparison of predictive performance and model reliability across all datasets and classification scenarios.

We selected Logistic Regression, Support Vector Machine, Decision Tree, and Random Forest classifiers for model development because these algorithms are well-established for datasets with a high feature-to-sample ratio, such as miRNA sequencing data. Classical ML algorithms provide stable performance in small cohorts and allow interpretable insights into feature contributions, which is essential in exploratory biomarker research. Although deep learning and transformer-based models have recently emerged in the literature, these architectures generally require large training datasets to avoid overfitting and are not suitable for our pilot-study setting. Therefore, we opted for classical ML methods that offer better reliability and generalizability for limited sample sizes [24].

Generative Artificial Intelligence (GenAI) tools were not used in the preparation, writing, or editing of this manuscript.

## 3. Results

The median age of patients with endometriosis was 27 years (range 18–43), compared with 27 years (range 20–36) for patients with adenomyosis. In the control group (Endometriosis), most patients (*n* = 29) were diagnosed with peritoneal endometriosis (#ENZIAN P), followed by ovarian (#ENZIAN O, *n* = 6) and tubal (#ENZIAN T, *n* = 2) endometriosis. Deep infiltrating endometriosis was identified in 17 patients. The characteristics of the control group (endometriosis) are summarized in Table 1.

### 3.1. Classification Performance

The table presents the accuracy of four classification models—Logistic Regression, Decision Tree, Random Forest, and SVM—evaluated across four comparison groups involving adenomyosis samples from serum and urine (Table 2).

In the serum dataset comparing adenomyosis to positive controls (Figure 1a), Logistic Regression, Random Forest, and SVM showed comparable performance, whereas the Decision Tree model performed less favorably with an accuracy of 0.71. When serum samples from adenomyosis patients were compared with negative controls (Figure 1b), all four models achieved an accuracy of 0.60.

For urine samples, overall performance was higher. In the comparison of adenomyosis versus positive controls (Figure 1c), Logistic Regression, Random Forest, and SVM each reached an accuracy of 0.88, while the Decision Tree model achieved 0.82. In the urine dataset comparing adenomyosis to negative controls (Figure 1d), performance improved further: Logistic Regression, Random Forest, and SVM each attained an accuracy of 0.90, and the Decision Tree model achieved 0.80.

Confusion matrix analyses confirmed the performance differences observed across biofluids. In the serum-based models, all classifiers showed high specificity but consistently misclassified adenomyosis cases, indicating limited sensitivity. In contrast, urine-derived models demonstrated clearly improved discrimination, correctly identifying most negative controls and a higher proportion of adenomyosis cases. When compared with positive controls, urine-based models maintained strong classification of controls, although some adenomyosis cases were still predicted incorrectly. Overall, these results support the superior diagnostic potential of urinary miRNA signatures relative to serum in this pilot cohort (Appendix A).

### 3.2. ROC

To evaluate the diagnostic performance of the classification models, receiver operating characteristic (ROC) curve analysis was performed for all serum and urine datasets comparing adenomyosis with control groups. The area under the curve (AUC) was used as a quantitative measure of each model’s discriminatory ability.

In the serum dataset comparing adenomyosis with negative controls, the best-performing models were Logistic Regression and SVM, both achieving an AUC of 0.76 (Figure 2a,b). Random Forest demonstrated moderate performance with an AUC of 0.71, whereas the Decision Tree model performed poorly with an AUC of 0.43, indicating limited generalizability in this setting.

When comparing serum adenomyosis samples with positive controls, Random Forest and Logistic Regression achieved high AUC values of 0.98 and 0.95, respectively, demonstrating excellent classification performance (Figure 2c,d). SVM also performed strongly with an AUC of 0.90, while the Decision Tree model again showed reduced predictive ability, achieving an AUC of only 0.56.

In the urine dataset comparing adenomyosis to negative controls, Logistic Regression and SVM both reached AUC values of 1.00 (Figure 3a,b). Although these results indicate excellent separation in this dataset, such perfect performance must be interpreted with caution, as the very small number of adenomyosis cases (*n* = 7) increases the likelihood of overfitting rather than reflecting true diagnostic accuracy. Random Forest also performed strongly, with an AUC of 0.81, while the Decision Tree model achieved an AUC of 0.76.

When comparing urine samples from adenomyosis patients with positive controls, Random Forest again reached an AUC of 1.00 (Figure 3c). However, this result should similarly be viewed as an artifact of the limited sample size rather than as evidence of perfect discriminatory power. SVM and Logistic Regression showed good classification performance, with AUCs of 0.88 (Figure 3d) and 0.86, respectively, whereas the Decision Tree model demonstrated poor accuracy, achieving an AUC of 0.50.

Taken together, the results suggest that Random Forest, Logistic Regression, and SVM perform best in distinguishing adenomyosis from control groups based on urine miRNA profiles, although validation in larger cohorts is essential to confirm these preliminary findings and reduce the risk of overfitting.

### 3.3. Selected Features (20 MiRNAs)

Following the final step of Recursive Feature Elimination, the top 20 most relevant miRNAs were identified for each classification task (Figure 4a–d). These features represent the most informative candidates for distinguishing adenomyosis from the respective control groups across both serum and urine samples.

Serum: Adenomyosis vs. Positive Control: mir-641, mir-17, miR-6716-5p, mir-550a-1, mir-550b-1, miR-142-5p, miR-1233-5p, mir-4453, mir-4446, mir-5186, mir-548d-2, mir-378a, mir-3132, miR-1908-5p, miR-99a-5p, miR-145-5p, mir-9901, mir-3648-2, mir-4523, mir-6753.

Serum: Adenomyosis vs. Negative Control: mir- 5186, mir-10a, mir-17, mir-3132, mir-200c, miR-98-5p, mir-4446, miR-548z, mir-486-2, mir-181b-2, mir-3180-3, mir-3180-1, mir-7150, mir-320d-2, mir-155, mir-31, miR-205-5p, mir-5685, mir-548z, mir-4766.

Urine: Adenomyosis vs. Positive Control: miR-503-5p, miR-183-3p, miR-3529-3p, miR-128-3p, miR-339-3p, miR-1260a, mir-9-1, mir-4509-1, mir-671, mir-9902-2, mir-653, mir-423, mir-4509-3, mir-4509-2, mir-744, miR-450a-5p, miR-8077, miR-424-3p, miR-3130-5p, mir-8077.

Urine: Adenomyosis vs. Negative Control: mir-6069, mir-6724-2, miR-183-3p, mir-331, mir-320b-2, mir-193b, mir-129-2, mir-320d-2, miR-519d-5p, miR-8077, mir-6894, mir-20b, mir-500a, mir-363, mir-224, mir-132, mir-222, mir-6089-2, mir-4472-1, miR-550a-5p.

In the serum datasets, miR-17, miR-3132, and miR-5186 showed the highest feature weights, consistent with the previously reported regulatory role of the miR-17 family in inflammation and proliferative signaling associated with adenomyosis [18]. In the urine datasets, miR-183-3p and miR-8077 demonstrated the strongest feature importance scores, indicating a consistent discriminatory contribution across both comparisons (adenomyosis vs. positive controls and adenomyosis vs. negative controls). These miRNAs were repeatedly selected across multiple algorithms and cross-validation folds, suggesting that their expression patterns were among the most stable and informative for distinguishing adenomyosis from endometriosis and from control samples. Notably, miR-320d-2 emerged as a top-ranked feature in both serum and urine when comparing adenomyosis to negative controls, suggesting its potential as a systemic biomarker detectable across biofluids and its relevance for differentiating adenomyosis from patients with chronic pelvic pain without endometriosis.

### 3.4. Heatmap of Selected miRNAs

To visualize the discriminatory power of the selected features, heatmaps of the top 20 miRNAs were generated for each comparison (Figure 5a–d). Across all datasets, the heatmaps reveal distinct expression patterns and clear clustering of adenomyosis samples relative to both positive and negative controls, indicating that the selected miRNA panels capture biologically meaningful differences. In the serum datasets, adenomyosis samples exhibited characteristic coordinated up- and down-regulation across several miRNAs (e.g., miR-17, miR-3132, miR-5186), forming clusters separate from endometriosis controls and suggesting disease-specific regulatory alterations. In the urine datasets, expression patterns showed even stronger segregation: adenomyosis samples clustered tightly together and consistently demonstrated elevated levels of miR-183-3p and miR-8077, whereas control samples displayed markedly different expression intensities. This pronounced separation is in line with the superior performance of urine-based models observed in the ROC analyses. Overall, the heatmaps illustrate that adenomyosis is associated with coherent, biofluid-specific miRNA expression profiles and confirm the strong discriminative capacity of the selected miRNA features.

## 4. Discussion

Based on our findings and previous evidence, this pilot study adds to the growing literature on the role of microRNAs in pathophysiology and potential non-invasive diagnosis of adenomyosis. The dysregulated miRNAs identified in both serum and urine samples are consistent with key pathological mechanisms of adenomyosis, including altered cell proliferation, invasion, migration, inflammatory signaling, and impaired apoptosis [30,31,32]. These pathways are increasingly recognized as central drivers of disease progression.

Our results align with prior studies demonstrating that miR-183 suppresses epithelial cell migration and invasion in adenomyosis by targeting MMP-9 [33]. In our urine datasets, miR-183-3p emerged as a consistently relevant biomarker in both comparisons (adenomyosis vs. positive controls and adenomyosis vs. negative controls), supporting its biological plausibility. MiR-8077 was also identified in both urine comparisons; although its role in adenomyosis has not yet been described, its recurrent selection suggests potential diagnostic relevance that warrants further investigation.

In serum, miR-17, miR-3132, miR-5186, and miR-4446 were among the top discriminative features. The regulatory role of miR-17 in adenomyosis, particularly via the lncRNA H19/miR-17/TLR4 pathway, has been previously established [34]. In contrast, miR-3132, miR-5186, and miR-4446 have not been previously linked to adenomyosis, suggesting they may represent novel candidate biomarkers. Notably, miR-320d-2 was detected among the top features in both serum and urine when comparing adenomyosis to negative controls. While this miRNA has not been associated with adenomyosis, members of the miR-320 family have been implicated in chronic pain–associated conditions such as interstitial cystitis [35]. Its presence across biofluids suggests the potential for a systemic marker capable of differentiating adenomyosis from other causes of pelvic pain.

Additional support for a complex regulatory landscape in adenomyosis comes from reports of co-dysregulated circular RNAs (circRNAs) in the eutopic endometrium and the endometrial–myometrial interface [36]. Together, these observations indicate that adenomyosis likely involves an interconnected network of non-coding RNAs, reinforcing the value of miRNA-based approaches.

Our machine-learning models demonstrated encouraging diagnostic performance, particularly for urine-derived miRNA profiles. Random Forest, Logistic Regression, and SVM consistently outperformed the Decision Tree model and showed high discriminatory ability across comparisons. However, the AUC values of 1.00 observed in several urine-based models must be interpreted with great caution. Given the very small number of adenomyosis-only cases (*n* = 7), such perfect results are likely due to overfitting rather than true diagnostic performance. Larger, independent cohorts are required to validate these findings and ensure model generalizability.

The limited sample size represents the main constraint of this pilot study and may have influenced both model behavior and the composition of the selected biomarker panels. Small datasets are inherently susceptible to instability in feature ranking, which may explain the absence of certain well-established miRNAs (e.g., miR-145) among the top-ranked markers. Moreover, the low number of adenomyosis-only patients prevented stratification by lesion extent or coexisting conditions. As such, the present study should be viewed as a proof of concept demonstrating methodological feasibility rather than a definitive diagnostic panel. Validation in larger, phenotypically well-characterized cohorts will be essential.

Despite these limitations, this study provides important methodological insights. The structured ML pipeline—comprising variance filtering, univariate testing, mutual information, and recursive feature elimination—allowed reduction of more than 4000 miRNAs to compact, high-performing marker sets. Such approaches are particularly valuable in high-dimensional, low-sample settings, where classical statistical methods often fail. ML-based models offer a powerful framework for detecting complex biomarker patterns that may otherwise remain undetected using traditional techniques [37,38,39]. Our findings reinforce the utility of ML-assisted analysis in uncovering subtle, biofluid-specific expression signatures associated with adenomyosis.

Given the substantial clinical overlap between adenomyosis and endometriosis, the ability of miRNA-based ML models to differentiate between these conditions holds potential clinical significance. Previous work by our group demonstrated that serum miRNAs can distinguish endometriosis from other causes of chronic pelvic pain [40]. In contrast, the current study identified a completely different miRNA pattern for adenomyosis, with no overlap among the top-ranked features. This supports the hypothesis that adenomyosis and endometriosis, although often co-occurring, may involve distinct regulatory mechanisms at the miRNA level.

Compared with imaging modalities such as TVUS and MRI, which have moderate sensitivity and specificity [7,8,9,10], miRNA-based approaches may serve as complementary tools to support earlier diagnosis, improve patient stratification, and guide fertility management. In particular, urine-derived miRNAs appear to be highly promising due to their strong discriminative performance and the ease of non-invasive sample collection. Nevertheless, these results should be considered hypothesis-generating, and larger multicenter cohorts—ideally incorporating deep learning and longitudinal validation—are required.

Recent work by Bendifallah et al. [21,24,25] has demonstrated the feasibility of salivary miRNA-based screening for endometriosis using ML models, achieving excellent diagnostic performance across development and validation cohorts. Our study extends this concept to adenomyosis and to additional biofluids (serum and urine), representing an important step toward multi-fluid, AI-assisted diagnostics for benign gynecological disorders.

To support reproducibility, all analyses in this study were performed using standardized and widely adopted pipelines, including nf-core/smRNAseq for sequencing data processing and DESeq2, scikit-learn, and Nextflow for statistical and machine-learning workflows. Although ethical restrictions prevent public release of raw sequencing data, processed datasets and code can be made available upon reasonable request. Replicability across centers will require larger adenomyosis cohorts, but the methodological framework is broadly transferable.

## 5. Conclusions

Adenomyosis is a clinically relevant condition that remains challenging to diagnose using imaging alone. The identification of reliable, non-invasive biomarkers is therefore an important objective, as earlier and more accurate detection may improve treatment selection and reproductive outcomes. This pilot study evaluated the diagnostic potential of serum- and urine-derived miRNA profiles using next-generation sequencing combined with machine-learning methods.

Our results show that adenomyosis is associated with distinct miRNA expression signatures in both biofluids, with urine-based models demonstrating the strongest discriminatory performance. Several miRNAs—such as miR-183-3p, miR-17, and miR-320d-2—emerged as promising candidates for differentiating adenomyosis from endometriosis and from negative controls, supporting the feasibility of miRNA-based liquid biopsy approaches for non-invasive adenomyosis diagnostics.

This study has important limitations, particularly the small number of adenomyosis-only cases and the lack of external validation, which increase the risk of overfitting and limit generalizability. Larger, multicenter cohorts are needed to confirm the stability of the identified signatures and to assess their clinical applicability. Future research should also explore the mechanistic relevance of these miRNAs and investigate whether integrating clinical parameters with miRNA data can further enhance diagnostic accuracy.

In summary, this pilot work provides preliminary evidence that serum and urine miRNA profiles may serve as non-invasive biomarkers for adenomyosis. These findings lay the groundwork for future validation studies and the development of biomarker-supported diagnostic tools in gynecologic practice.

## Figures and Tables

**Figure 1 diagnostics-15-03012-f001:**
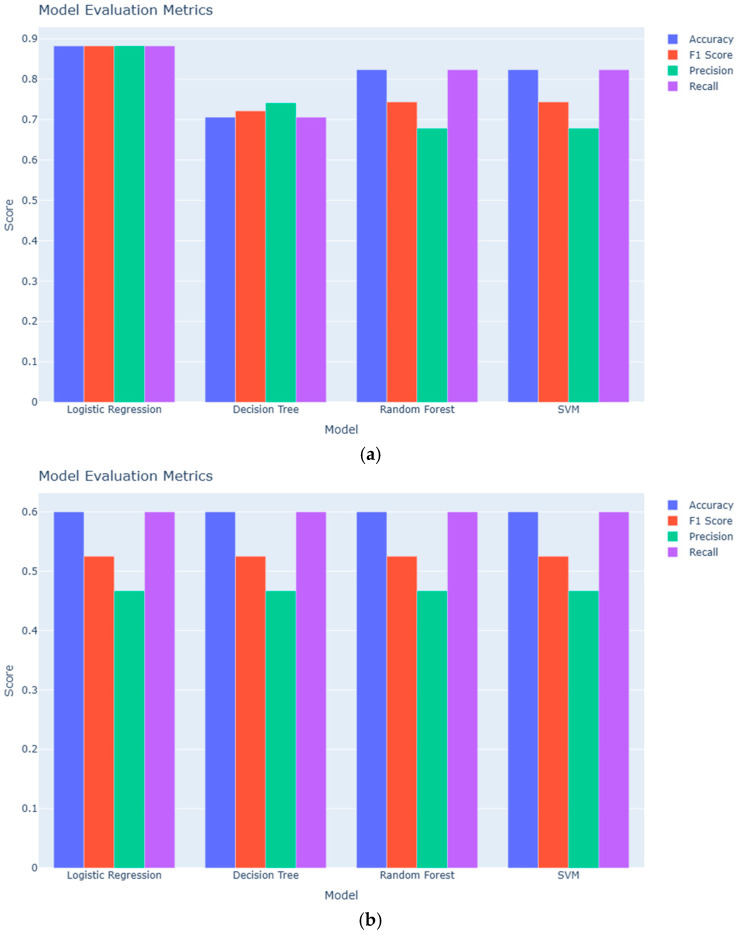
Performance comparison of machine learning models applied to serum and urine miRNA data. Bar chart showing accuracy, F1 score, precision, and recall for four models: Logistic Regression, Decision Tree, Random Forest, and SVM. (**a**) Serum: Adenomyosis vs. positive controls. (**b**) Serum: Adenomyosis vs. negative controls. (**c**) Urine: Adenomyosis vs. positive controls. (**d**) Urine: Adenomyosis vs. negative controls.

**Figure 2 diagnostics-15-03012-f002:**
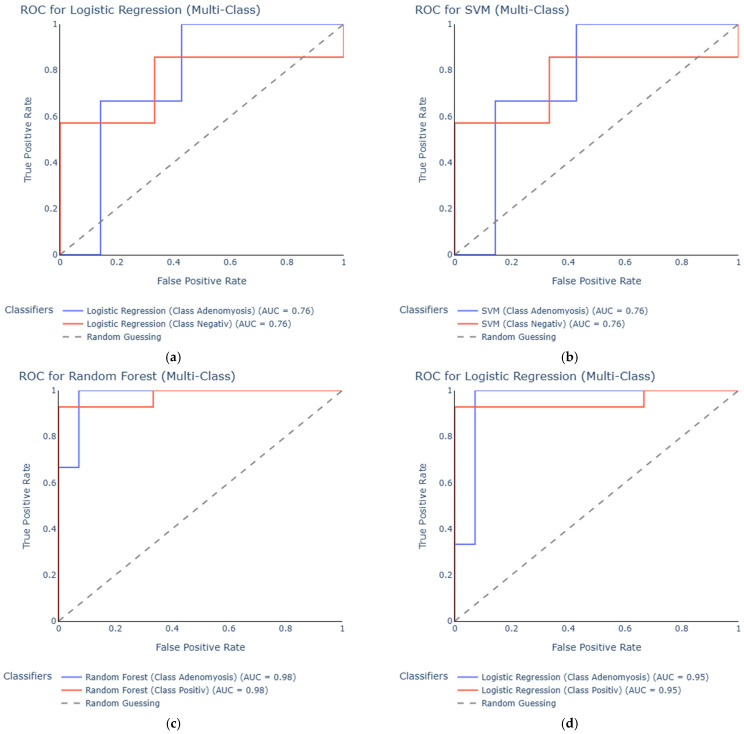
ROC curves for serum-based classification models comparing adenomyosis with control groups. The curves show the best performing models logistic regression (**a**) and SVM (**b**) for adenomyosis vs. negative controls and random forest (**c**) and logistic regression (**d**) for adenomyosis vs. positive controls.

**Figure 3 diagnostics-15-03012-f003:**
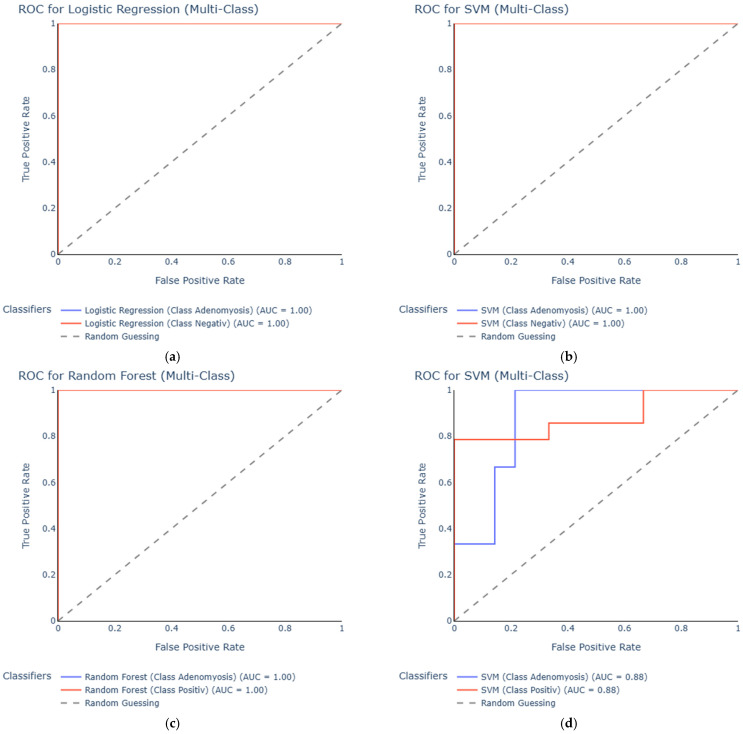
ROC curves for the urine dataset. Comparing adenomyosis vs. negative controls, the best performing models were logistic regression (**a**) and SVM (**b**). In the adenomyosis vs. positive controls dataset the best performing models were random forest (**c**) and SVM (**d**).

**Figure 4 diagnostics-15-03012-f004:**
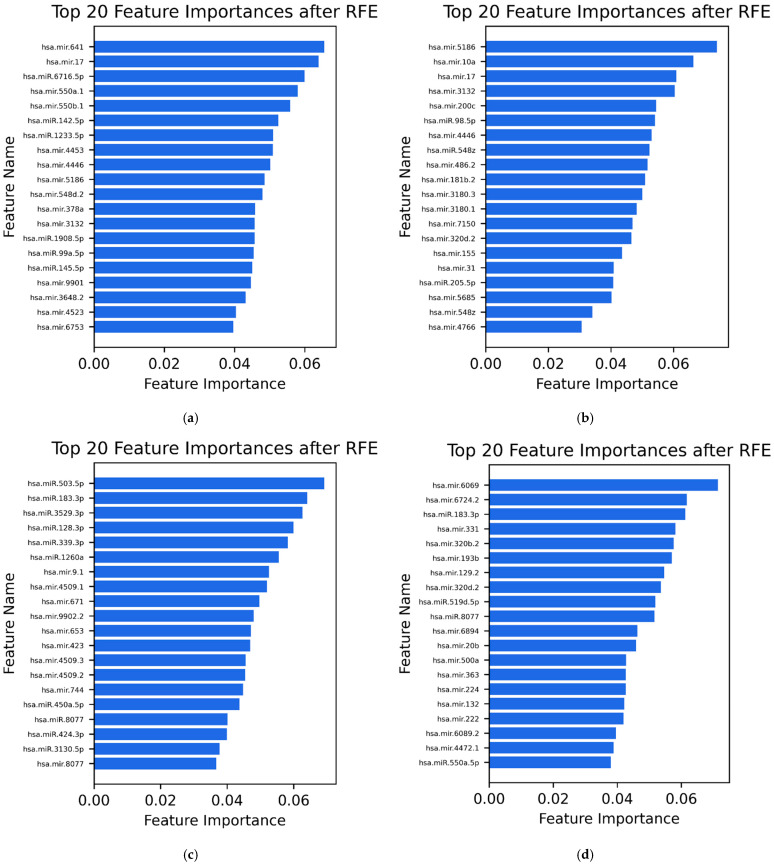
Top 20 most informative miRNAs for the classification of adenomyosis based on serum and urine samples, identified using RFE. (**a**) Serum: Adenomyosis vs. positive controls. (**b**) Serum: Adenomyosis vs. negative controls. (**c**) Urine: Adenomyosis vs. positive controls. (**d**) Urine: Adenomyosis vs. negative controls.

**Figure 5 diagnostics-15-03012-f005:**
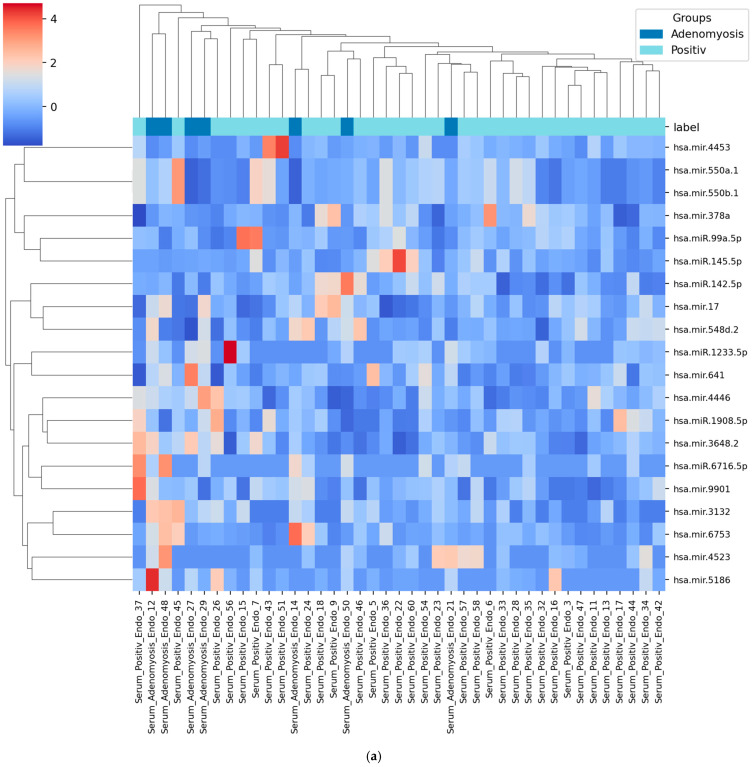
Heatmaps of the top 20 selected miRNAs illustrating expression differences across groups. Each panel represents the expression profiles of the top miRNAs in serum or urine samples, stratified by disease status. Clear clustering patterns support the discriminative power of selected miRNAs. (**a**) Serum: Adenomyosis vs. positive controls. (**b**) Serum: Adenomyosis vs. negative controls. (**c**) Urine: Adenomyosis vs. positive controls. (**d**) Urine: Adenomyosis vs. negative controls.

**Table 1 diagnostics-15-03012-t001:** Characteristics of patients with endometriosis (control group).

	Median (Range)
Age	27 (18–34)
Location of Endometriosis (#ENZIAN)	Number of Patients *n*
#ENZIAN P (Peritoneal)	29
#ENZIAN O (Ovary)	6
#ENZIAN T (Tube)	2
Deep infiltrating endometriosis	17
#ENZIAN A	5
#ENZIAN B	12
#ENZIAN C	0
#ENZIAN FB	2
#ENZIAN FI	1

**Table 2 diagnostics-15-03012-t002:** Accuracy of different classification models used in the evaluation of the data set.

Model	Accuracy (Serum: Adenomyosis vs. Positive Controls)	Accuracy (Serum: Adenomyosis vs. Negative Controls)	Accuracy (Urine: Adenomyosis vs. Positive Controls)	Accuracy (Urine: Adenomyosis vs. Negative Controls)
Logistic Regression	0.88	0.60	0.88	0.90
Decision Tree	0.71	0.60	0.82	0.80
Random Forest	0.82	0.60	0.88	0.90
SVM	0.82	0.60	0.88	0.90

## Data Availability

The original contributions presented in this study are included in the article/Appendix A. Further inquiries can be directed to the corresponding author.

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
