# Peer review of "Urine and Serum miRNA Signatures for the Non-Invasive Diagnosis of Adenomyosis: A Machine Learning-Based Pilot Study"

_diagnostics, 2025, doi:10.3390/diagnostics15233012_

Round 1

Reviewer 1 Report

Comments and Suggestions for Authors

Thank you for the opportunity to comment on the paper by Kupec et al., “Urine and Serum miRNA Signatures for the Non-Invasive Diagnosis of Adenomyosis Uteri: A Machine Learning-Based Pilot Study.”

General comment:

- I appreciate this study - it reflects a lot of work and enthusiasm, and an intention to address a real clinical need. However, in its current form, the superficial treatment of several parts of the manuscript reduces its credibility and readability, so a substantial revision is necessary.

- The translational significance of miRNAs in adenomyois is up-to-date, the diagnostic implications are potentially clinically relevant. I congratulate mdpi for publishing on these research gaps in adenomyosis - these days, an exhaustive systematic review on non-coding RNAs (miRNAs, lncRNAs, circRNAs) in adenomyosis was published in IJMS, also an MDPI journal (doi: 10.3390/ijms262110713). This paper provides an appropriate translational context also for the submitted manuscript and should be carefully studied by the authors.

- Acknowledging that the study was done: the manuscript is written in a very uneven way and each part (Introduction, Methods, Results, Discussion) requires revision.

Language and style

- The language is awkward and several sentences are grammatically incorrect. For instance: “samples with a pseudonym” (should be “pseudonymized samples”), “a urine samples are taken” (singular/plural), “in the endometriosis center” (better “at the…”), “7 ml of cell-free supernatant was available, 4 ml was used…” (should be “were available / were used”), “All samples from 34 positive controls, 7 adenomyosis uteri and 18 negative controls” (should be “7 patients with adenomyosis” or “7 adenomyosis cases”), and “Location of endometriosis diagnosis” (probably “Location of endometriosis”).

- The frequent use of the Latin/German term “adenomyosis uteri” should be replaced by “adenomyosis.”

A) Introduction

The research context is missing. The authors mention in one sentence the role of miRNAs in pathological processes in general, but they do not explain the role of miRNAs in adenomyosis at all (!). The systematic review cited above would provide an excellent background.

B) Methods

1) For TVUS diagnosis of adenomyosis, the widely used and consensus-based term “Morphological Uterus Sonographic Assessment (MUSA) criteria" should be applied instead of only citing “Harmsen et al. [6]”.

2) According to the journal’s instructions, the authors must eventually “disclose in this section details of how GenAI has been used in the paper use of GenAI” - the declaration about “what was not used/done”, as provided here, makes no sense.

C) Results

1) Given the very small adenomyosis group (n = 7), reporting median with range or IQR would be more appropriate than mean ± SD in the characteristics table.

2) Figure 4, which presents the key message of the paper, is hardly legible.

3) the same applies to Figure 5 (heat maps).

4) The “feature importance” is not exposed at all in the narrative results.

D) Discussion

1) In the discussion, the authors mainly explain how machine learning can be applied to biomarker discovery (which is already well established), but they devote only one sentence without any reference (“Several miRNAs identified in our study (e.g., miR-183, miR-17) are consistent with previous reports…”)  to what has been achieved in adenomyosis miRNA research so far: The proportions of the discussion should be changed: please discuss in more detail the context of miRNAs in adenomyosis and shorten the part about machine learning.

2) Some sentences are not logically connected, for example, the statement that ML has been applied to interpret large-scale data is followed by “In this context, our findings contribute…”, although the connection is not shown.

3) The use of abbreviations vs. full terms is inconsistent ( “machine learning” appears after “ML”).

4) The very small sample of only 7 adenomyosis patients is not explained/justified (were no more adenomyosis patients identifiable in a tertiary endometriosis center?).

5) The impact of this limited sample size on the results is not discussed, although it may influence the unexpected composition of the potential miRNA signature (e.g. absence of miR-145). No further limitations are discussed.

6) It is also unclear why the potential of miRNA signatures for the differential diagnosis against endometriosis is not discussed, even though endometriosis patients dominated the control group. This is particularly surprising because the same authors have recently published on this topic (“Diagnostic Potential of Serum Circulating miRNAs for Endometriosis in Patients with Chronic Pelvic Pain”, doi: 10.3390/jcm14145154).

7) The series of relevant works by Bendifallah et al. is not specifically discussed, but instead mentioned only in a general sentence on “non-invasive applications” of miRNAs.

E) References

The reference format does not follow the MDPI style and needs to be adapted.

Comments on the Quality of English Language

A careful language editing is mandatory.

Reviewer 2 Report

Comments and Suggestions for Authors

This article presents the use of machine learning algorithms using non-invasive biomarkers of serum and urine microRNA (miRNA) profiles for the diagnosis of adenomyosis uteri. Various feature selection algorithms were applied to the data, with the highest success rate achieved with the Random Forest algorithm.

1. The sentence containing ref. [23] should be checked. Similarly, the last sentence of the first paragraph of section 2.1 is missing ".". The article needs to be edited.

2. The biggest problem with such studies is the lack of generalizability of the results. The dataset is very small. There are 59 data in total. Incorrectly classifying a patient during the testing process results in an error of approximately 5%.

3. The ROC curves indicate that the learning process was insufficient due to the insufficient dataset.

4. Accuracy and error changes should be provided to demonstrate that the training process was completed correctly. Accuracy and error changes should have similar changes for the validation and training data.

5. Confusion matrices should be included in the article.

6. Generally, images are blurry. Clearer images should be provided.

Comments on the Quality of English Language

There are some spelling errors.

Reviewer 3 Report

Comments and Suggestions for Authors

1- You forgot to change the content in the “Data Availability Statement” section. You should revisit that section.

2- I believe the total number of patients in the study is low at 59. The number of patients in the study must be increased to boost confidence in the generalization capabilities of the proposed models.

3- Logistic Regression, Support Vector Machine, and Random Forest seem a bit outdated to me as methods. Current science is moving towards deep learning-based, even transformer-based systems. Frankly, the authors should consider working on newer architectures.

4- It is stated that the study data was collected between December 2021 and August 2023. Frankly, more than 24 months have passed since the last data collection. You should consider adding new data to your dataset.

5- In the “2.5 Model Selection and Evaluation” section, you should write the reasons for the models you selected. You should explain in detail why you did not use other models in the literature.

6- You should consider presenting the obtained AUC results in a separate table.

7- The generated heatmaps have not been explained at all. The results obtained in these graphs should be explained in a discussable manner. Researchers should consider students reading a new article in the field in their manuscripts and present not only the results but also their meanings.

8- Conclusions should be revisited in the manuscript. In modern research articles, this section begins with at least one paragraph of text that includes the importance and purpose of the topic. Subsequently, the results obtained are summarized and interpreted. Then, the importance of the results obtained is discussed. Often, the limitations of the study and future work are introduced in a separate paragraph. Authors should consider restructuring the Conclusions section in a standardized format.

9- A discussion regarding the reproducibility and replicability of the study should be added to the content. The use of the dataset and the visibility of the codes can be emphasized at this stage.

10- If possible, a time analysis should be added to the study, and the training time of the architecture should be provided within the study. Differences in time analysis between architectures should be compared.

11- You should discuss at length in the discussion section of your manuscript the contributions of the machine learning method to your results. The main contribution of your work is not apparent in its current form.

Round 2

Reviewer 1 Report

Comments and Suggestions for Authors

I appreciate the authors’ effort to improve the manuscript. This effort have resulted in a thoroughly revised, well-organized, and well-written manuscript that I can recommend for rapid publication. Excellent work!

Author Response

Reviewer comment:
I appreciate the authors’ effort to improve the manuscript. This effort have resulted in a thoroughly revised, well-organized, and well-written manuscript that I can recommend for rapid publication. Excellent work!

Response:
We sincerely thank the reviewer for this very positive assessment and greatly appreciate the supportive feedback. We are pleased that the revisions have improved the clarity, organization, and overall quality of the manuscript.

Reviewer 2 Report

Comments and Suggestions for Authors

Accuracy and loss change indicate whether there was overfitting during the training process. Accuracy and loss change were not included in the article. 

Author Response

Reviewer comment:
Accuracy and loss change indicate whether there was overfitting during the training process. Accuracy and loss change were not included in the article. 

Response:

We thank the reviewer for this important comment. Unfortunately, the training history (epoch-wise accuracy and loss change) was not stored during model development, and therefore cannot be retrospectively included. However, to address the concern regarding potential overfitting, we now provide detailed performance metrics for the held-out test set (Supplementary Table S2), including accuracy, precision, recall, and F1 score across all four classifiers

Reviewer 3 Report

Comments and Suggestions for Authors

I would like to thank the authors for carefully answering all the questions. As a reviewer, I am pleased to have been able to help the article reach a higher level. Thank you.

Author Response

Reveiwer comment:
I would like to thank the authors for carefully answering all the questions. As a reviewer, I am pleased to have been able to help the article reach a higher level. Thank you.

Response:
We sincerely thank the reviewer for the thoughtful and constructive feedback throughout the revision process. We greatly appreciate your valuable contribution, which has helped us further improve the quality and clarity of the manuscript. Thank you for your time and support.